# Callus Irradiation Is an Effective Tool for Creating New Seashore Paspalum Germplasm for Stress Tolerance

Zhuyi Bai [1], Qing Yu [1], Yuying Zheng [1], Zhiyong Wang [2], Yu Liu [3], Jun Liu [1], Zhimin Yang [1] and Yu Chen [1,*]

1 College of Agro-Grassland Science, Nanjing Agricultural University, Nanjing 210095, China
2 College of Forestry, Hainan University, Haikou 570228, China
3 College of Landscape Architecture, Jiangsu Vocational College of Agriculture and Forestry, Zhenjiang 212499, China
* Correspondence: cyu801027@njau.edu.cn

**Abstract:** Seashore paspalum (*Paspalum vaginatum* Swartz) is a perennial warm-season turfgrass which is known for its superb salinity tolerance. Compared to bermudagrass, seashore paspalum exhibited the adverse character of faster vertical growth, wider leaf, weak cold-, drought- and disease-resistance. In this study, we aimed to improve these unfavorable traits of seashore paspalum through the strategy of callus irradiation. The results showed that 2108 regenerated plants were obtained following the method of the seashore paspalum calluses irradiated by $^{60}$Co-$\gamma$ rays (dose: 60 Gy, dose rate: 1 Gy/min). Morphological traits were measured combining with cluster analysis on the regenerated plants to select mutant lines with short leaves (A24 and A82) and thin leaves (A24, A83, and A120) as well as dwarfism (B73, B28, B3, A29, and B74). In addition, we found various mutant characters such as greenish leaf sheath (A69 and A71), soft leaf (B77, B17, and B110), and strong erectness (B5 and B9) under continuous observation. Through the comprehensive tolerance analysis following the index of survive rate, relative water content, leaf electrolyte leakage, MDA content; photochemical efficiency and leaf wilting coefficient, three drought-tolerant lines (A55, B72, and B44) and one cold-tolerant line (B59) were screened. This research proved that callus irradiation is an effective way to create new seashore paspalum germplasm, which provides valuable materials for accelerating the breeding process of seashore paspalum and further excavating the molecular regulatory mechanisms of these traits in turfgrass.

**Keywords:** seashore paspalum; $^{60}$Co-$\gamma$ ray; mutant; drought tolerance; cold tolerance

## 1. Introduction

Seashore paspalum (*Paspalum vaginatum* Swartz) is a perennial warm-season turfgrass of *Gramineae*, which has excellent characteristics such as trampling resistance, close mowing tolerance, salt tolerance, and heavy metal tolerance. It is widely used in landscaping and ecological remediation [1]. However, compared to bermudagrass, seashore paspalum exhibited the adverse character of faster vertical growth, wider leaf, weak cold-, drought- and disease-resistance, which influencing the turfgrass quality and greatly limiting its application in arid and cold areas in winter [2–8]. The improvement of these characters has always been the focus of seashore paspalum breeding.

The technology of $^{60}$Co-$\gamma$ ray radiation mutagenesis has been widely used in turfgrass breeding, and related varieties have been reported on bermudagrass (*Cynodon dactylon*), centipedegrass (*Eremochloa ophiuroides*), and other grass species [9–13]. Zhang et al. [14] irradiated the stolon of bermudagrass with $^{60}$Co-$\gamma$ rays and obtained 12 mutants with high turf quality and low inflorescence density. These mutants were confirmed differences at the DNA level. Lu et al. [15] used $^{60}$Co-$\gamma$ rays to irradiate the stolons of bermudagrass, and 3 lines with dwarfism and greater drought tolerance were obtained. Chen et al. [16] irradiated the seeds of common bermudagrass with $^{60}$Co-$\gamma$ rays, screened a dwarf-type mutant (S-20-1) with greater drought tolerance, and found that the dwarfism of S-20-1

might be due to the decrease of GA$_3$ or insensitivity to GA$_3$. Lu et al. [17] screened three mutant lines (22-1, 22-2, 22-3) by irradiating the seeds of centipedegrass with [60]Co-γ rays. Mutant 22-2 showed obvious dwarfism, while mutants 22-1 and 22-3 had no obvious change in morphological characteristics, but the cold and drought tolerance was improved. Subsequently, it was found that the improvement of stress resistance of the two mutants was associated with their high levels of polyamines (PAs) and antioxidant defense system [18,19]. Li et al. [20] irradiated the stem segments and calluses of St. Augustinegrass (*Stenotaphrum secundatum*) with [60]Co-γ rays, 13 morphological mutants were selected from mutagenized mutants, and most of the mutants showed semi-dwarfism.

In the study of [60]Co-γ irradiation mutation of seashore paspalum, stolon was mainly used as irradiation treatment material. In general, the suitable irradiation dose of seashore paspalum stolon was 40–70 Gy, and the survival rate of stolon was negatively correlated with the radiation dose. Nine mutant lines with excellent characteristics were selected from the irradiation mutants, including three with high drought tolerance and one with excellent salt tolerance [21–24]. In the aspect of the radiation mutagenesis of seashore paspalum callus, only the screening of irradiation dose has been reported. Ye et al. [25] irradiated the embryogenic calluses of seashore paspalum Adalay with [60]Co-γ rays to study the effects of different irradiation doses on callus differentiation and plant regeneration. 60 Gy was screened as the suitable dose for calluses of seashore paspalum, but the screening of related mutants has not been reported. To sum up, the application of [60]Co-γ ray radiation mutagenesis technology is used in seashore paspalum (Table 1). In the aspect of chemical mutagenesis of seashore paspalum callus, the application of colchicine and ethyl methanesulfonate (EMS) has been deeply studied and breeding results have been achieved [26–29]. This work aims to obtain a large number of mutant lines by [60]Co-γ ray induced callus mutagenesis and enrich seashore paspalum germplasm resources, further screening the excellent morphological and stress tolerant materials. The implementation of this study will provide excellent parental materials for breeders, accelerate the target breeding process, and establish a basis for further positive mining of trait genes.

**Table 1.** The application of [60]Co-γ ray radiation mutagenesis on seashore paspalum.

| Species | Varieties | Materials | Mutagen | Irradiation Parameters | Results | Reference |
|---------|-----------|-----------|---------|------------------------|---------|-----------|
| *Paspalum vaginatum* | Adalay | Callus | [60]Co-γ | Dose: 0, 20, 30, 40, 50, 60, 70, 80 Gy<br>Dose rate: 1 Gy/min | 60 Gy is the suitable irradiation dose for callus. At this dose, the rate of shoot regeneration was 76.13%, and the rate of plant regeneration was 41.94%. | [25] |
| | Platinum, Sea Isle 2000 | Stolon | [60]Co-γ | Dose: 0, 40, 50, 60 Gy<br>Dose rate: 0.12 Gy/min | The suitable irradiation dose for Platinum was 41.38–49.75 Gy. The suitable irradiation dose for Sea Isle 2000 was 54.07–64.89 Gy. | [24] |
| | Sea Isle 2000, Platinum, Supreme, Salam | Stolon | [60]Co-γ | Dose: 0, 40, 45, 50, 55 Gy<br>Dose rate: 0.12 Gy/min | Nine mutant lines with excellent characteristics were selected from the irradiation mutants of four varieties. | [21] |

## 2. Materials and Methods

### 2.1. Plant Materials

The experiment material is the callus of seashore paspalum (*Paspalum vaginatum* Swartz) Sea Spray. The calluses were induced from erect stems and subcultured for many generations, and all the erect stems were obtained from one pot.

### 2.2. Creation of Mutant Materials

Two batches of callus (line A and line B) were irradiated in November 2020 and April 2021. After 25 days of subculturing, the calluses were irradiated by [60]Co-γ rays at a dose

of 60 Gy and a dose rate of 1 Gy/min [25]. After irradiation, the calluses were recovered in the dark (line A recovered for 5 days, line B recovered for 12 days) and transferred onto a differentiated medium, and the medium was changed every 30 days until root induction and further development. The plantlets regenerated from the irradiation were transplanted to plastic pots with the soil mixture of peat and vermiculite (7:3, *v/v*). After being maintained in the greenhouse for two months, the plantlets were transferred to the resource garden for maintenance.

The data used for statistical analysis are as follows: survival rate (%) = (number of surviving callus/number of irradiated callus) × 100, differentiation rate (%) = (number of shoot callus / number of irradiated callus) × 100, relative differentiation rate (%) = (number of shoot callus / number of surviving callus) × 100, relative plant regeneration rate (%) = (number of regenerated plant / number of shoot) × 100.

### 2.3. Study on Morphological Variation of Mutants

In August 2021, 200 mutant lines and WT (wild type, regenerated from unirradiated callus) with coverage of more than 95% were selected and mowed to the same height. The leaf length, leaf width, and plant height were measured on the 20th day after mowing. The leaf length is the distance from the tip of the leaf to the ligule, the leaf width is the distance from the widest part of the leaf, and both leaf length and leaf width were measured by the antepenult leaf of the erect stem, and five replicates for each line. The plant height is the height of potted materials in their natural state. Meanwhile, the rest of the phenotypic mutants were observed continuously.

### 2.4. Screening and Evaluation of Drought-Tolerant Mutants

2.4.1. Screening of Drought-Tolerant Mutants

In September 2021, cuttings (with two apical nodes and three leaves) collected from erect stems of 186 mutant lines and WT were cut into 50-cell plug trays. There are five replicates for each line, which followed a completely randomized design. Plants were allowed for growing 42 days in the greenhouse, with routine management by twice-weekly irrigation, biweekly mowing at 5 cm, and weekly fertilizing with 28-6-12 N-P-K fertilizer (Miracle-Gro). For drought treatment, each tray was irrigated with 3000 mL of water, followed by withholding irrigation for 20 days in the greenhouse. The plant wilting coefficient was used to evaluate the plants that were still alive in plugs.

The plant wilting coefficient indicates the wilting degree of the whole plant under drought stress, which can be described as seven grades: Grade 1, the plant status when WT dried out and died, scored as 0; Grade 2, the plant showed a severe wilting status, scored 10 points; Grade 3, the status of plant showed wilting, scored 10 points; Grade 4, the status of plant showed the beginning of wilting, scored 15 points; Grade 5, a large number of plant leaves yellowed; scored 20 points; Grade 6, the plant leaves began to yellow, scored 25 points; Grade 7, the plant showed no obvious signs of stress, scored 30 points.

2.4.2. Evaluation of Drought Tolerance of Mutants

In November 2021, six selected drought-tolerant lines and WT were planted in PVC tubes (11 cm diameter, 25 cm long) filled with sand, with 20 erect stems per tube and five replicates for each line. Plants were cultivated until coverage of more than 95% in the greenhouse, with an average temperature of 25/20 °C (day/night), a 14/10 h light/dark photoperiod, and routine management of plants as described above. For drought treatment, plants were fully irrigated, followed by withholding irrigation for 18 d and rehydration for 7 d in the greenhouse. During the drought treatment, the plant position was changed daily, and samples were taken at 0 d, 6 d, 12 d, and 18 d. The following physiological indexes were measured.

The leaf relative water content (RWC) was calculated as $[(FW - DW)/(TW - DW)] \times 100$, FW is leaf fresh weight immediately after harvesting, where TW is leaf turgid weight after soaking in deionized water for 24 h, and DW is leaf dry weight following oven-drying

for 96 h at 80 °C. The soil volumetric water content (SWC) was measured by the time domain reflectometry method (MiniTrase, Soilmoisture Equipment Crop., Goleta, CA, USA). The leaf electrolyte leakage (EL) was calculated as $(C_0/C_1) \times 100$. The conductivity of the solution ($C_0$) was measured using a conductivity meter (Orion Star A212, Thermo Fisher Scientific Inc., Waltham, MA, USA). Leaves were then killed by autoclaving at 121 °C for 20 min and cooled to room temperature, and the conductivity of killed tissues ($C_1$) was measured. For determination of MDA content, the content was determined by using a spectrophotometer, and the concentration of MDA was calculated using a coefficient of absorbance of 155 $mM^{-1} \cdot m^{-1}$. RWC, SWC, EL, and MDA were determined as described previously [30]. The maximum photochemical efficiency of PS II (Fv/Fm) in leaves was measured by Chlorophyll Fluorometer (OS30p+, Opti-Sciences, Inc., Hudson, NY, USA). The wilting degree of the whole leaf under drought stress was indicated by the leaf wilting coefficient (LWC) as described previously [22]. The comprehensive evaluation of drought tolerance was indicated by membership function, and D value represents the comprehensive evaluation value of drought tolerance [31].

### 2.5. Evaluation of Cold Tolerance of Mutants

Since 20 November 2021, Nanjing began to be affected by the cold wave, with rainfall and a sharp drop in temperature. The minimum temperature on the morning of 23–24 November was 0 to −2 °C [32]. On the 26th, line B59 was found to remain in a good state after exposure to the cold wave, and then its cold tolerance was evaluated.

B59 and WT were propagated in plots, with 15 replicates per line, and 15 erect stems per replicate. Plants were cultivated until coverage of more than 95% in the greenhouse, growing environment, and routine management of plants as described above. For cold treatment, plants were first transferred to a plant growth chamber (ZRX-1100G, Saifu experimental instrument Co., Ltd., Ningbo, China) for a week at 28 °C, then gradually exposed to 4 °C (with lowering 2 °C per hour) for 3 days of low-temperature acclimation, followed by further lowered the temperature to −7 °C (at a rate of 1 °C per hour) for cold treatment [26]. The treatment times were 4, 6, 8, 10, and 12 h, with three replicates at each time. After cold treatment, plants were thawed at 4 °C for 24 h, then the EL of leaves was measured.

### 2.6. Statistical Analysis

All data were subjected to analysis of variances using IBM SPSS Statistics (Version 25). Duncan's t-test was used to evaluate differences among treatments and plant lines at 0.05 probability level. For morphological traits, cluster analysis was done with OriginPro (Version 2021) using the Euclidean distance matrix in the Hierarchical cluster analysis method, normal distribution was tested using the Kolmogorov-Smirnov (K-S) test, and morphological mutants were screened using the box and whisker plots (boxplots) [33,34]. In our analysis, the fences of the boxplot are defined as $Q1 - 0.55 \times IQR$ and $Q3 + 0.55 \times IQR$, where $Q1$ is the lower quartile, $Q3$ is the upper quartile, and $IQR$ is the interquartile range.

## 3. Results

### 3.1. Effect of Irradiation on Callus of Seashore Paspalum

Irradiation significantly inhibited the survival rate, differentiation rate, and plant regeneration rate of callus. After irradiation, the survival rate of callus was only 35.69%. The differentiation rates of line A and line B were 17.46 and 32.99%, respectively, the relative differentiation rates were 56.48 and 89.12%, respectively, and the relative plant regeneration rates were about 27.41% (Table 2). After transferring into differentiated medium, most of the calluses could not further differentiate and induce shoot with browning and vitrification. Among the differentiated shoots, a large number of shoots showed stasis phenomenon and even aborted (Figure 1). Finally, 2108 mutant lines were obtained.

**Table 2.** Plantlet differentiation and plant regeneration of irradiated callus.

|  | Line A | Line B | Total | Control |
|---|---|---|---|---|
| No. of irradiated callus | 1672 | 6035 | 7707 | 115 |
| No. of surviving callus | 517 | 2234 | 2751 | 111 |
| No. of shoot callus | 292 | 1991 | 2283 | 109 |
| No. of shoot | 497 | 7193 | 7690 |  |
| No. of regenerated plant | 121 | 1987 | 2108 | 106 |
| Survival rate (%) | 30.92 | 37.02 | 35.69 | 96.52 |
| Differentiation rate (%) | 17.46 | 32.99 | 29.62 | 94.78 |
| Relative differentiation rate (%) | 56.48 | 89.12 | 82.99 | 98.20 |
| Relative plant regeneration rate (%) | 24.35 | 27.62 | 27.41 |  |

Note: Line A is the first batch of irradiated calluses; line B is the second batch of irradiated calluses; Control is the unirradiated calluses.

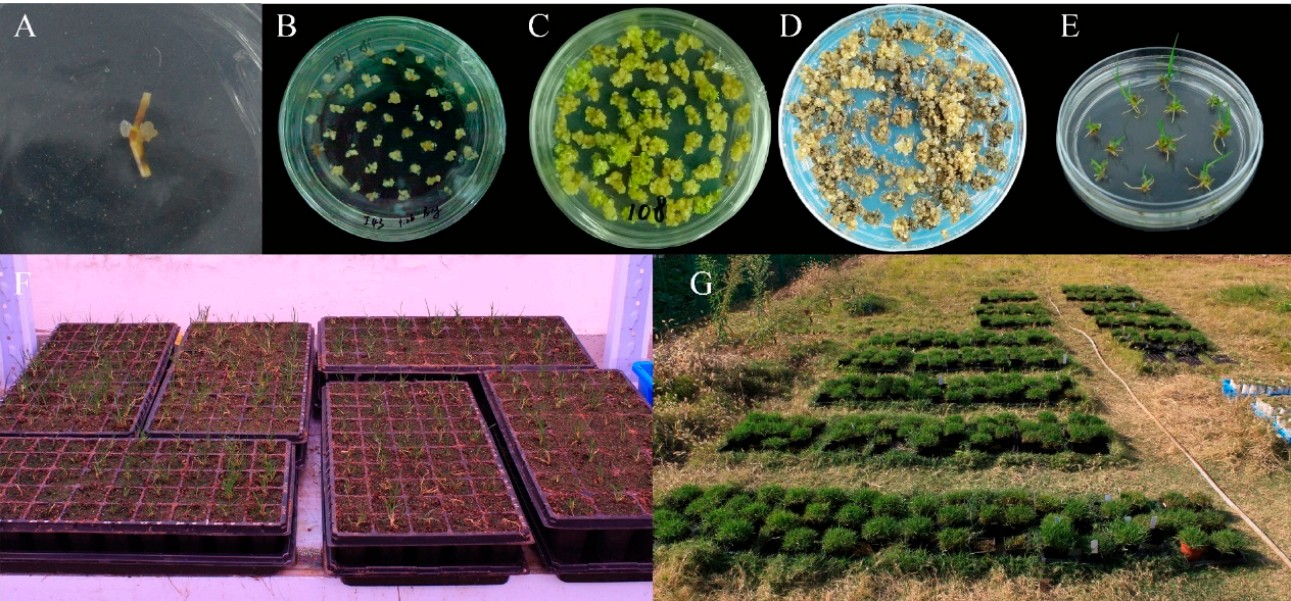

**Figure 1.** The regeneration of callus induced by $^{60}$Co-γ ray. (**A**) Calluses induced from stem segment; (**B**) calluses before irradiation; (**C**) calluses with shoots; (**D**) dead calluses after irradiation; (**E**) regenerated plantlets on differentiated medium; (**F**) regenerated plantlets in the greenhouse; (**G**) regenerated plants in the resource garden.

*3.2. Morphological Variation of Mutants*

For leaf length, 201 lines showed a normal distribution, and the coefficient of variation was 32.29%. At Euclidean distance 1.6, the mutant lines were divided into three leaf length types: long-leaf type (16 lines), general type (88 lines, WT belonged to this type, same below), and short-leaf type (97 lines), and the short-leaf type accounted for 48.26% (Figure 2A,B). The results indicated that irradiation induced a high frequency of variation in leaf length, and the main variation was shortened leaf length. Based on the results of cluster analysis, lines beyond fences of boxplot were screened as mutant lines using boxplot (Figure 2C). Sixteen long-leaf mutant lines were obtained: B17 (6.23 cm), B14 (6.42 cm), B61 (6.62 cm), B52 (6.67 cm), B46 (6.74 cm), B54 (7.07 cm), B33 (7.10 cm), B158 (7.30 cm), B68 (7.41 cm), B49 (7.45 cm), B128 (7.55 cm), B79 (7.62 cm), B31 (7.78 cm), B56 (8.25 cm), B50 (8.25 cm), B65 (8.29 cm). 2 short-leaf mutant lines: A24 (2.09 cm), A82 (2.07 cm).

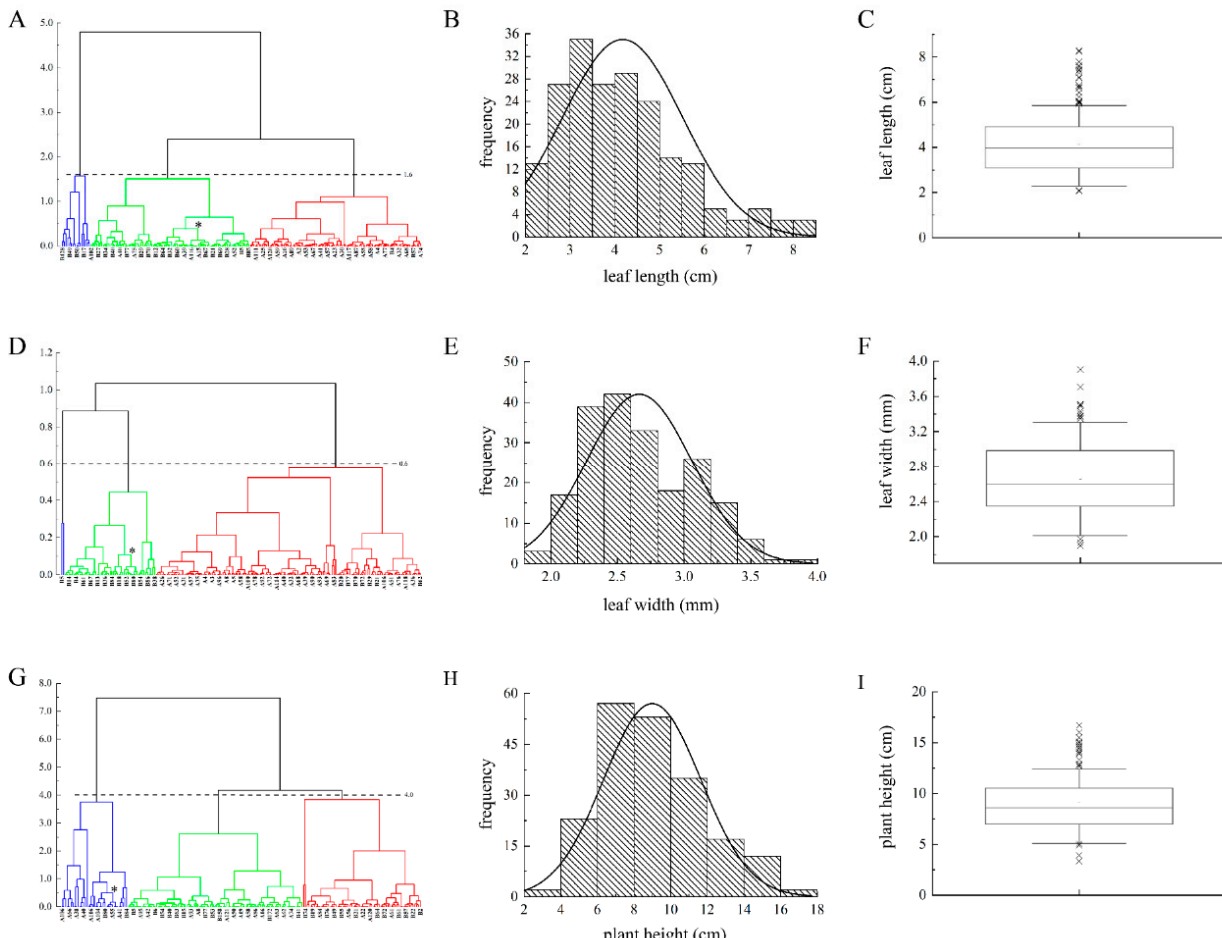

**Figure 2.** An analysis of leaf length, leaf width, and plant height of 201 materials. (**A**) The dendrogram of leaf length; (**B**) the frequency distribution histogram of leaf length; (**C**) the box plot of leaf length; (**D**) the dendrogram of leaf width; (**E**) the frequency distribution histogram of leaf width; (**F**) the box plot of leaf width; (**G**) the dendrogram of plant height; (**H**) the frequency distribution histogram of plant height; (**I**) the box plot of plant height; * indicates that WT is in this subtree; the overlapped labels are hidden in dendrogram; × indicates outliers.

For leaf width, 201 lines showed a normal distribution, and the coefficient of variation was 14.89%. At Euclidean distance 0.6, the mutant lines were divided into three leaf width types: broad-leaf type (2 lines), general type (52 lines), and thin-leaf type (147 lines), and the thin-leaf type accounted for 81.09% (Figure 2D,E). The results indicated that the variation of leaf width of mutant lines showed a reduction in leaf width. Based on the results of cluster analysis, 2 broad-leaf mutant lines: B87 (3.71 mm), B5 (3.90 mm); and 3 thin-leaf mutant lines: A24 (1.97 mm), A83 (1.93 mm), and A120 (1.90 mm) were screened by boxplot (Figure 2F).

For plant height, 201 lines showed a normal distribution, and the coefficient of variation was 29.70%. At Euclidean distance 4.0, the mutant lines were divided into three plant height types: general type (38 lines), semi-dwarf type (97 lines), and dwarf type (66 lines), and the dwarfism accounted for 72.13% (Figure 2G,H). The results indicated that irradiation induced a high frequency of variation in plant height, and the majority of the phenotypes were reduced plant height. As shown in Figure 2I, 5 significant dwarf lines were screened by boxplot: B73 (5.05 cm), B28 (4.90 cm), B3 (4.90 cm), A29 (3.90 cm), and B74 (3.35 cm).

*3.3. Observation of Phenotypic Mutants*

There were abundant phenotypic variations in the obtained mutant lines, such as dwarfism (B74, A29), thin-leaf (B172), broad-leaf (B5), greenish leaf sheath (A69 and A71)

(Figure 3), soft leaf (B77, B17, and B110), strong erectness (B5 and B9) and lighter leaf color. Among those mutants, we were particularly interested in four lines, B30, B74, B5, and B172 (Figure 4); B30 has a similar phenotype to WT, but more dwarfed; B74 was dwarfism and grew faster than A29; B5 had strong erectness (fast erect growth) and stolons were not yet observed in the pot; B172 had significantly thinner leaves.

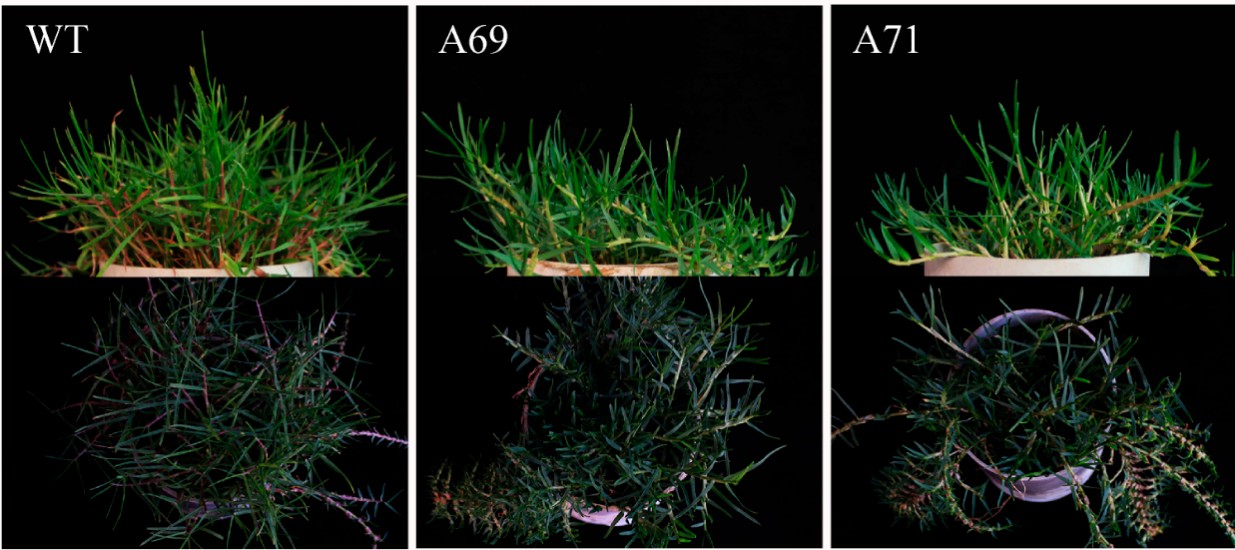

**Figure 3.** WT and mutants with a green leaf sheath color (A69 and A71).

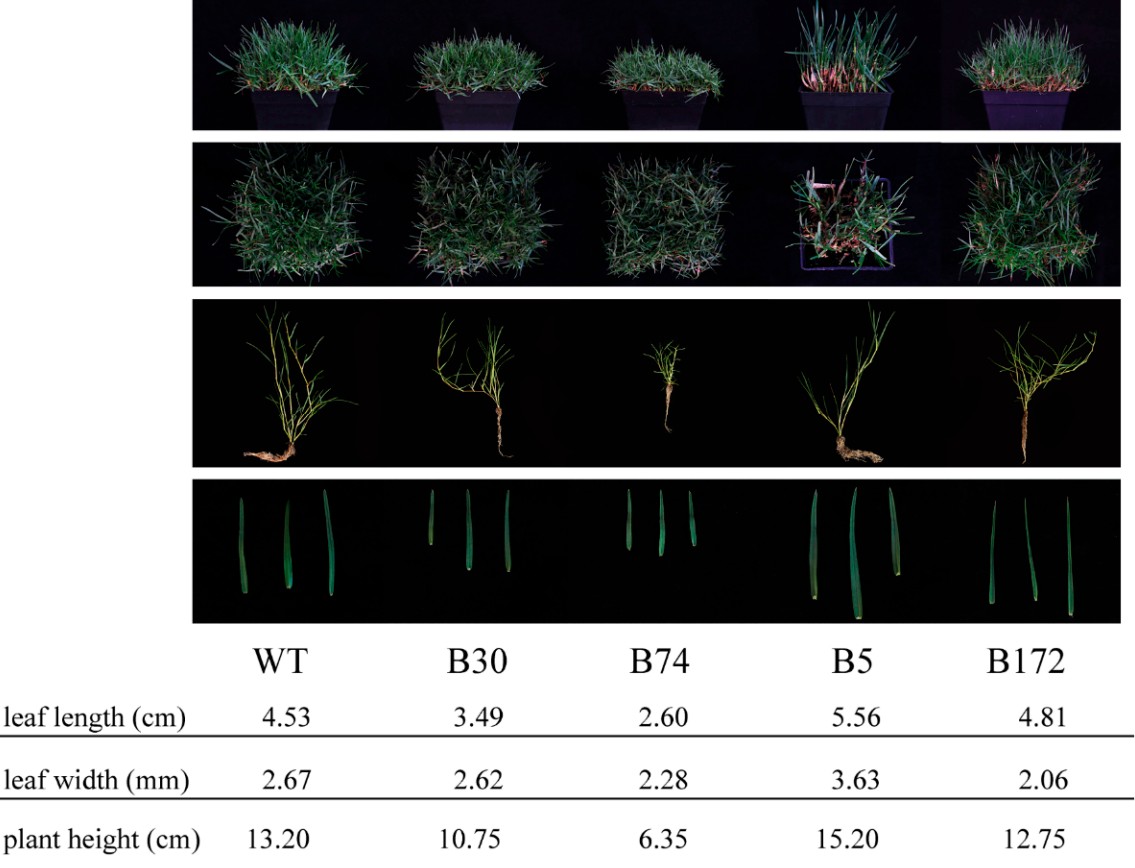

|  | WT | B30 | B74 | B5 | B172 |
|---|---|---|---|---|---|
| leaf length (cm) | 4.53 | 3.49 | 2.60 | 5.56 | 4.81 |
| leaf width (mm) | 2.67 | 2.62 | 2.28 | 3.63 | 2.06 |
| plant height (cm) | 13.20 | 10.75 | 6.35 | 15.20 | 12.75 |

**Figure 4.** WT and 4 representative mutant lines.

### 3.4. Screening of Drought-Tolerant Mutants

Only 76 lines had some replicates survived after drought treatment, among which only eight lines had more than three replicated were alive (Table 3). The plant wilting coefficient of surviving lines was scored, and six mutant lines (B44, A55, B82, A86, B72, and B2) were considered as drought-tolerant mutants.

**Table 3.** The scoring results of screening drought-tolerant mutans.

| Mutant Strain | Replicate 1 | Replicate 2 | Replicate 3 | Replicate 4 | Replicate 5 | Score |
|---|---|---|---|---|---|---|
| B44 | 25 | 25 | 25 | 20 | | 19 |
| A55 | 20 | 15 | 15 | 15 | | 13 |
| B82 | 25 | 15 | 15 | | | 11 |
| A86 | 25 | 20 | 5 | | | 10 |
| B72 | 20 | 15 | 15 | | | 10 |
| B2 | 20 | 15 | 15 | | | 10 |
| B5 | 10 | 10 | 5 | | | 5 |
| B162 | 5 | 5 | 5 | | | 3 |

Note: This table lists only lines with surviving replicates $\geq 3$ among the 76 lines.

### 3.5. Evaluation of Drought Tolerance of Mutants

The RWC of 6 mutant lines was significantly higher than WT at 6 and 12 d of drought (Figure 5A). At 18 d, the RWC of B72 (77.93%), B44 (75.77%) and A55 (74.96%) were similar to the RWC of WT (74.05%), and higher than B82 (64.94%), A86 (66.71%) and B2 (66.43%) (Figure 5A). The EL of mutant lines showed an increase followed by a decrease on 6 and 12 d of drought, while the WT increased continuously (Figure 5B). At 18 d, the EL of A55 (26.04%) and B44 (33.15%) was significantly lower than WT (43.11%) (Figure 5B). The MDA content of mutant lines and WT showed an increase followed by a decrease on 6 and 12 d of drought. At 18 d, the content of A55 (7.14 nmol·g$^{-1}$ FW) was significantly lower than other lines. B72 (9.04 nmol·g$^{-1}$ FW), B2 (9.71 nmol·g$^{-1}$ FW), and B44 (10.77 nmol·g$^{-1}$ FW) were lower than WT (11.65 nmol·g$^{-1}$ FW), but not significant (Figure 5C). There were differences in Fv/Fm between mutant lines and WT at 0 d, 6 d, and 12 d of drought, but there was no significant decrease overall, and the mean values of Fv/Fm were 0.792, 0.796, and 0.788, respectively. Then a significant decrease occurred at 18 d (Figure 5D). The leaf wilting coefficient of mutant lines and WT did not change at 6 d of drought, and both began to increase at 12 d. At 18 d of drought, the leaf wilting coefficient of A55 (2.44) and B72 (3.68) was significantly lower than other lines, and the leaf wilting coefficient of B44 (3.80) was lower than 4.00, which also performed better than WT (4.60) (Figure 5E). The SWC of mutant lines and WT maintained the same trend during the treatment (Figure 5F).

The LWC, SWC, Fv/Fm, RWC, EL, and MDA content of six mutant lines and WT were comprehensively evaluated. The D value indicated the comprehensive evaluation value of drought tolerance, and the larger the D value, the better the drought tolerance of the line. The D values were ranked from large to small as A55 > B72 > B44 > WT > B2 > A86 > B82, indicating that the drought tolerance of A55, B72, and B44 was better than WT (Table 4), which was consistent with the phenotype (Figure 6).

**Table 4.** A comprehensive evaluation of drought tolerance of 7 materials.

| Material | Subordinate Function Value | | | | | | D Value | Rank |
|---|---|---|---|---|---|---|---|---|
| | $X_1$ | $X_2$ | $X_3$ | $X_4$ | $X_5$ | $X_6$ | | |
| WT | 0.100 | 0.110 | 0.673 | 0.425 | 0.192 | 0.254 | 0.224 | 4 |
| B44 | 0.300 | 0.108 | 0.741 | 0.468 | 0.502 | 0.330 | 0.342 | 3 |
| A55 | 0.640 | 0.081 | 0.670 | 0.447 | 0.723 | 0.645 | 0.503 | 1 |
| B82 | 0.100 | 0.071 | 0.453 | 0.196 | 0.288 | 0.196 | 0.176 | 7 |

**Table 4.** *Cont.*

| Material | Subordinate Function Value | | | | | | D Value | Rank |
|---|---|---|---|---|---|---|---|---|
| | $X_1$ | $X_2$ | $X_3$ | $X_4$ | $X_5$ | $X_6$ | | |
| A86 | 0.160 | 0.082 | 0.701 | 0.240 | 0.209 | 0.163 | 0.189 | 6 |
| B72 | 0.330 | 0.076 | 0.757 | 0.522 | 0.387 | 0.481 | 0.362 | 2 |
| B2 | 0.070 | 0.034 | 0.491 | 0.233 | 0.261 | 0.422 | 0.213 | 5 |
| Weight | 0.201 | 0.228 | 0.050 | 0.137 | 0.169 | 0.216 | | |

Note: $X_1$, $X_2$, $X_3$, $X_4$, $X_5$, $X_6$ indicate the leaf wilting coefficient, soil water content, maximum photochemical efficiency of PS II, leaf relative water content, leaf relative electrical conductivity, and MDA content, respectively; D value indicates the comprehensive evaluation value of drought tolerance of each material.

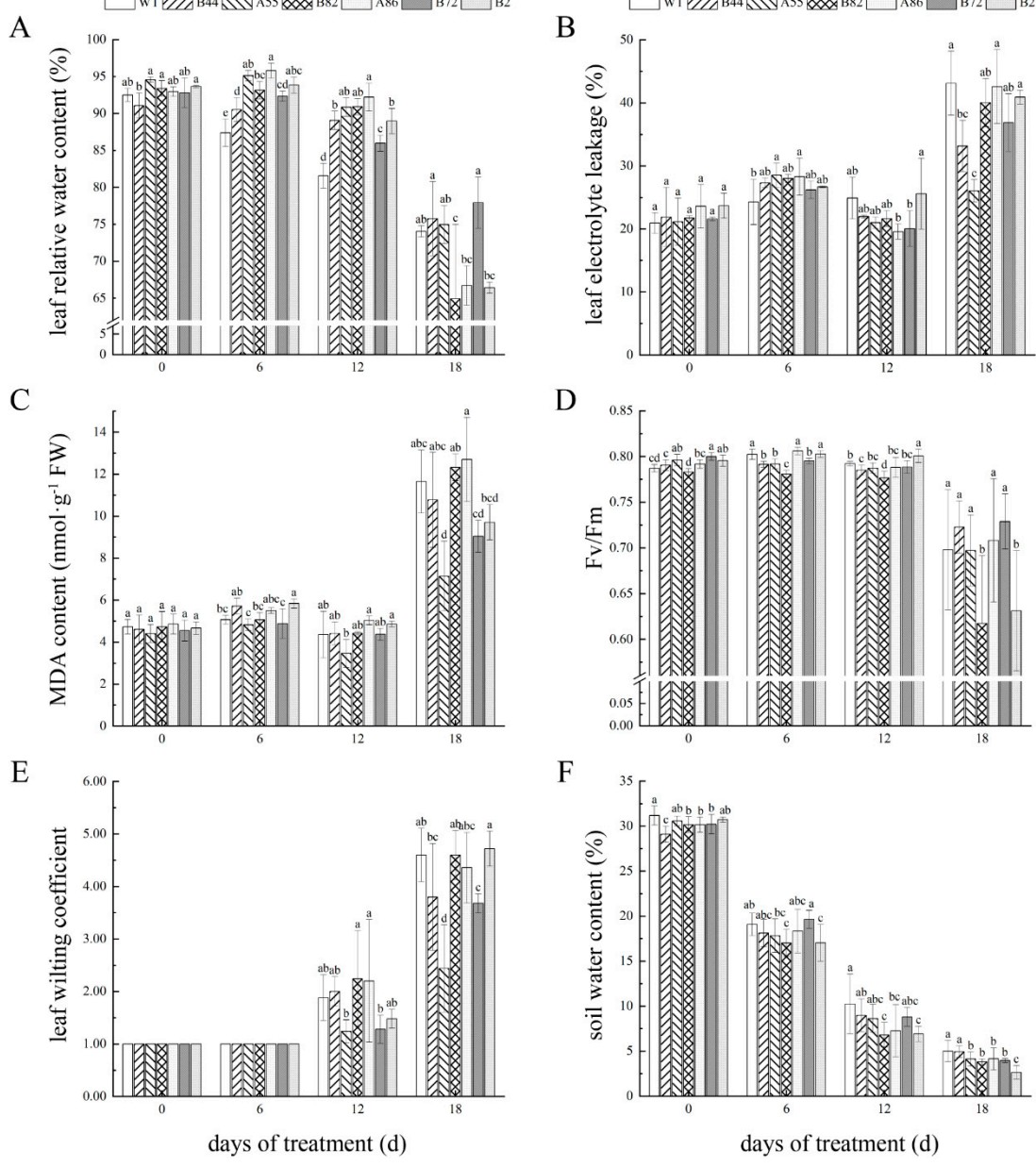

**Figure 5.** The changes of 6 physiological indexes under drought stress. (**A**) Leaf relative water content (RWC); (**B**) leaf electrolyte leakage (EL); (**C**) MDA content; (**D**) maximum photochemical efficiency of PS II (Fv/Fm); (**E**) leaf wilting coefficient; (**F**) soil water content (SWC); different lowercase letters in the same column indicate significant differences ($p < 0.05$).

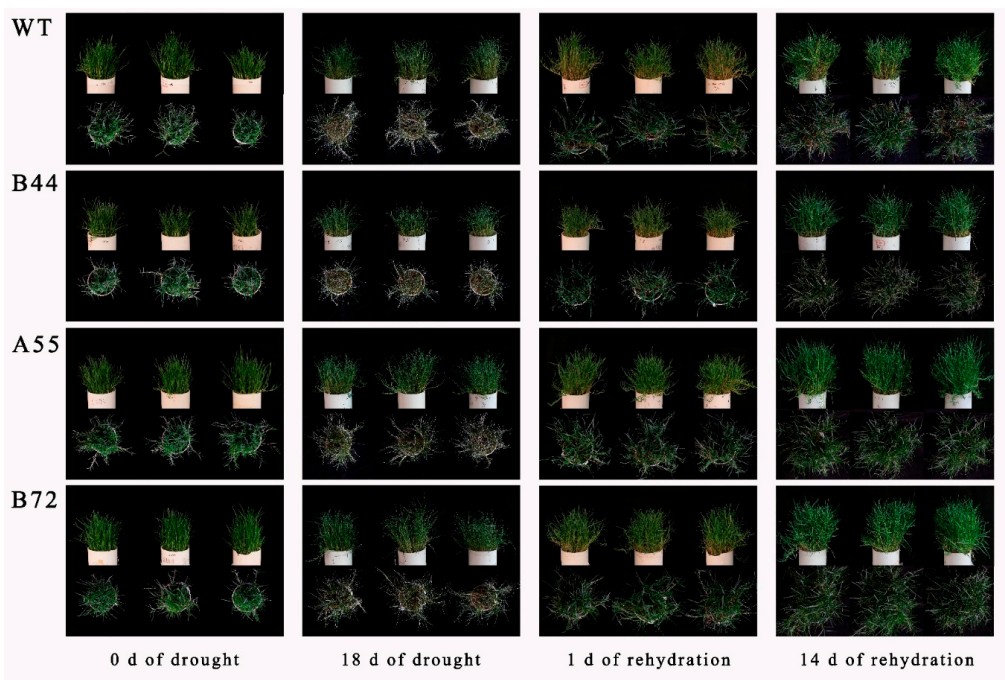

**Figure 6.** The performance of WT and three excellent drought-resistant materials at different stages of treatment.

### 3.6. Evaluation of Cold Tolerance of Mutants

The EL of WT exposed to −7 °C for 12 h was significantly higher than that of B59, which was consistent with the phenotype (Figure 7A–F). After cold treatment at −7 °C for 12 h, WT was severely damaged after 24 h thawing, with wilted and blackened leaves, yellowed leaf sheaths, and weakened stem segments. The performance of B59 was different, but there were differences among replicates. Replicate b maintained a better performance and was not affected. The performance of replicates a and c was the same as WT basically, but there were still some leaves and stem segments that maintained good mechanics and color and replicate a performed better.

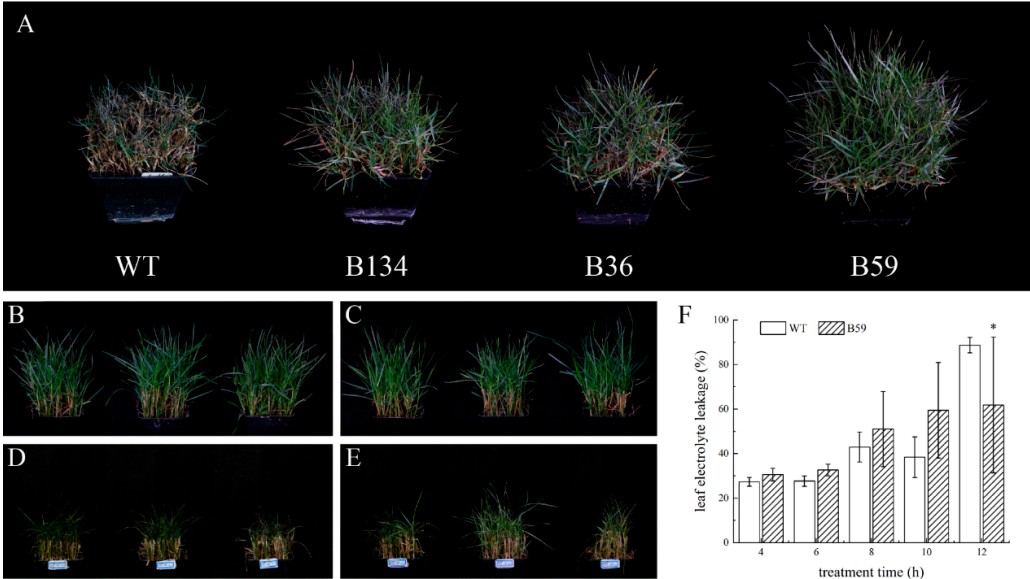

**Figure 7.** The performance of WT and B59 after freezing treatment. (**A**) Materials that perform well after the cold wave; (**B**) WT at 28 °C before freezing treatment; (C) B59 at 28 °C before freezing treatment;

(**D**) WT thawed at 4 °C for 24 h after freezing treatment; (**E**) B59 thawed at 4 °C for 24 h after freezing treatment; (**F**) Changes of electrolyte leakage in leaves after treatment at −7 °C for different time; * indicates significant differences ($p < 0.05$).

## 4. Discussion

### 4.1. Effect of Irradiation on Callus

The application of [60]Co-γ ray radiation in turfgrass mainly uses seeds and stem segments as irradiation treatment materials [10–13,16,35], the use of callus is less reported. Li et al., [20] screened three mutant lines with significant morphological changes using [60]Co-γ irradiated calluses of St. Augustinegrass. Lee et al., [36] used [60]Co-γ rays to irradiate the calluses of *Zoysia japonica*. It was found that 50–70 Gy was the appropriate dose range to induce zoysiagrass mutants, and three glyphosate-tolerant lines were screened. Chen et al. [37] discussed the irradiation effect of [60]Co-γ rays on the callus of *Zoysia matrella* and screened four salt-tolerant lines, and Lin et al. [38] screened one dwarf line in the follow-up observation. At present, the use of [60]Co-γ ray to irradiate callus of seashore paspalum has only been reported on irradiation dose screening, and this study found that the suitable irradiation dose for callus of Adalay is 60 Gy [25]. Since our primary goal was to obtain mutant lines, and both Adalay and Sea Spray are an identical species, we directly used 60 Gy as the dose of irradiation. In our experiment, the callus of seashore paspalum 'Sea Spray' was irradiated, the survival rate was 35.69%, and the differentiation rate was 29.62%, which was lower than the previous results (Table 1), indicating that the effect of the same dose and dose rate on calluses of different varieties of the same species were different. Meanwhile, two batches of calluses line A and line B with different recovery times after irradiation had different differentiation rates and relative differentiation rates, indicating that the effect of irradiation on the differentiation ability of callus was related to the recovery time of calluses after irradiation [39,40].

### 4.2. Morphological Variation of Mutants

After radiation mutation, seashore paspalum can obtain phenotypic variation materials such as shortened stolon internodes, thinning of leaves, and dwarfism of plants. When large number of rich variation materials were generated, the screening work is tedious [21,24]. Sen and Alikamanoglu [41] classified 10 sugar beet (*Beta vulgaris)* mutants and their parents into two distinct clusters based on ISSR markers. With the same method, Khatri et al., [42] divided the banana mutant lines into three clusters and two groups. The above is a classification of materials based on the results of molecular markers. In terms of morphological traits, Amri–Tiliouine et al. [43] grouped 135 $M_2$ chickpea (*Cicer arietinum*) lines in nine clusters using cluster analysis of phenotypic performance in ten quantitative traits. Atay et al. [44] divided 815 apple mutants and four commercial cultivars into six clusters using cluster analysis on six architectural traits. In this experiment, cluster analysis was carried out on leaf length, leaf width, and plant height, and mutant lines were classified into three types based on different traits, respectively. Short-leaf type, thin-leaf type, and dwarfism accounted for the majority of the corresponding traits, indicating that the use of radiation mutation is more likely to obtain mutants with short leaves and dwarf plants, which is in line with the breeding needs of turfgrass [45]. Meanwhile, plant growth was damaged by irradiation, and negative growth of plants with dwarfism as a typical phenotype was also found in other plants [46–48].

The phenotyping of mutant lines using cluster analysis narrowed the scope of screening, but the mutant population in this study was still rather large. Ambati et al. [49] used boxplots to analyze the variation of eight quantitative traits in 604 durum wheat (*Triticum aestivum*), for breeding high quality durum wheat. The boxplot analysis can not only be used to analyze the degree of variation but also can be applied to screen materials. Liu et al., [33,50] discussed the method of screening example varieties by boxplot in the DUS test of *Tagetes erecta*, and applied it in the DUS test of *Ranunculus asiaticus*. Peng et al. [34] screened 31 mutant lines from five physiological indexes of *Lolium perenne* by boxplot. In

this experiment, the leaf length, leaf width, and plant height of mutant lines were tested by normality test and cluster analysis. Combined with the results of cluster analysis, two short-leaf lines, three thin-leaf lines, and five significantly dwarf lines were screened.

In the results of the normality test, the distributions of leaf length, leaf width, and plant height were all right-skewed, which means that the data were more clustered on the left side of the distribution, while the tail in the positive direction is longer. There are similar distributions in quantitative traits of species, such as *Lolium multiflorum* [51], soybean [52], *Citrus* [53], *Canarium album* [54], and Chinese cherry [55]. In comparison with the results of cluster analysis, some of the outliers in the positive direction also belong to the general type (which WT belongs to) of cluster analysis, which was caused by the right-skewed distribution of the quantitative traits. We combined cluster analysis, normality test, and boxplot analysis to get the final screening result.

### 4.3. Screening and Evaluation of Drought-Tolerant and Cold-Tolerant Mutants

In turfgrass mutation breeding, resistance mutants were usually screened in long-term observation [15–17,19,22,56–58]. Applying corresponding abiotic stress at different stages of plant growth can effectively improve screening efficiency. Liu et al., [59] treated the embryogenic calluses of seashore paspalum at low temperature and obtained cold-tolerant mutants at 2 or 6 °C for 90 days. Xu et al., [60] screened aseptic propagation buds of Baxijiao (*Musa acuminata*) induced by EMS with L-HYP directed screening. The surviving aseptic seedlings performed well under low-temperature stress and cold-tolerant mutants were obtained. For the screening of drought-tolerant mutants, the use of PEG solution to simulate drought stress can easily control the variables, requires less treatment time, and can quickly screen the target materials [41,61–63]. However, there are essential differences between osmotic stress and natural drought. In this experiment, the erect stem cuttings were cultured in plug tray and treated with natural drought. According to our experience, plants planted in the plug tray can be treated in about 40 days, which shortens the culture time of the plant. At the same time, the volume of plug tray is small, and the soil layer of the matrix is shallow, so the plant responds quickly under stress. When all replicates of WT dried out and died, the alive mutant lines were evaluated and six drought-tolerant lines were screened.

The drought tolerance of the above six lines was evaluated. In the face of drought stress, maintaining a high RWC is very important for plants to continue their metabolic activities and maintain normal physiological and biochemical functions [64]. Under stress, a large number of oxygen radicals are produced in plants, and membrane lipid peroxidation is intensified which increases the production of MDA, which destroys the membrane structure system, changes the composition of membrane proteins, increases the permeability of plasma membrane, and exudates a large number of electrolytes and other content substances [65]. In this experiment, the RWC of WT under drought stress decreased first and was always lower than the RWC of A55, B72, and B44. At 18 days of drought, the EL of A55 and B44 was significantly lower than the EL of WT. The MDA content of A55 and B72 was significantly lower than that of WT. The results showed that the plasma membrane structure of the three drought-tolerant lines was more stable and the water-holding capacity of leaves was stronger. The drought tolerance of six mutant lines and WT was comprehensively evaluated by membership function, and three drought-tolerant mutant lines (A55, B72, and B44) were screened. The growth rate of A55 and B72 was similar to WT, while the growth rate of B44 was not only slower than WT, but also obviously different from WT in phenotype, especially the thickening, hardening, and obvious leathering of leaves.

In addition, we found a mutant line B59 with good cold tolerance among the potted lines in the field and continued to identify its cold tolerance. Shi et al. [26] identified a cold-tolerant mutant line A6 through the temperature resulted in 50% electrolyte leakage (TEL$_{50}$) screen in regenerated seashore paspalum plantlets obtained by EMS, and treated them at low temperature from 5 to −7 °C. The results showed that the expression of DREB1/CBFs

was changed and CAT activity and free amino acid concentrations were increased in mutant line A6 under cold stress. In this experiment, the EL of B59 was significantly lower than WT when treated at −7 °C for 12 h, and the phenotype of B59 after thawing was also better than WT, so it was considered that the cold tolerance of B59 was better than WT.

## 5. Conclusions

The use of whole plantlet or part of plant organs and tissues as irradiation treatment materials is generally considered to be prone to chimera. In contrast, the use of callus as irradiation treatment material is theoretically a single cell mutation, which is conducive to the acquisition of stable mutants, and shortens the breeding cycle. Our results proved that $^{60}$Co-γ ray irradiation on callus is an effective method to create new germplasms of seashore paspalum. With this method, we firstly obtained many mutant lines including two short-leaf lines, three thin-leaf lines, five significantly dwarf lines, three drought-tolerant lines, and one cold-tolerant line. In addition, we also found several interesting mutated characters such as greenish leaf sheath, soft leaf, and strong erectness. At present, all the mutant lines have been transplanted to the field for further experiments. After many generations of propagation, the morphological variation lines are genetically stable, and the lines with abiotic stress tolerance are under continuous observation. This research can provide novel materials for the breeding of domestic seashore paspalum varieties and also lay the foundation for excavating the molecular regulatory mechanisms of these morphological and stress tolerance traits in seashore paspalum or other turfgrass.

**Author Contributions:** Y.C. and Z.Y. conceived the study and designed the experiments. Z.B., Q.Y. and Y.Z. performed the experiments. Z.B. and J.L. analyzed the data with suggestions by Y.C. and Z.W.; Z.B. and Y.C. wrote the manuscript and Y.L. revised the manuscript. All authors have read and agreed to the published version of the manuscript.

**Funding:** This work was supported by the funding program (number 31872953, 31672193) of National Natural Science Foundation of China NSFC.

**Institutional Review Board Statement:** Not applicable.

**Informed Consent Statement:** Not applicable.

**Conflicts of Interest:** The authors declare no conflict of interest.

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
