# Peer review of "Callus Irradiation Is an Effective Tool for Creating New Seashore Paspalum Germplasm for Stress Tolerance"

_agronomy, doi:10.3390/agronomy12102408_

Round 1

Reviewer 1 Report

Paspalum vaginatum, a perennial turfgrass, has useful characteristic for landscaping and ecological remediation, but it also has disadvantages like weak drought tolerance and cold tolerance. The authors irradiated the calluses of P. vaginatum with 60Co-γ rays and screened the mutant lines with higher drought- and cold-tolerance.

The screening processes, the characterization of regenerated plants, and the examination of drought- and cold-tolerance appear to be fine. However, for readers who would like to repeat the experiments, some of the most critical data are missing.

For example, the authors describe that the gamma-ray dose (60 Gy) was determined based on the previous work, which was written in Chinese and most people would not obtain information easily. Also, the authors mention that the survival rate and the differentiation rate were lower than the previous results due to the difference of variety, but do not provide detailed information on the extent of these differences. The authors should state such information clearly at the beginning of the result about how they determined the irradiation dose and what were the results, and what was the difference with the previous work. One option would be to summarize the data in a table, for example.

After looking at the survival and the regeneration rates shown in this work, the more optimal dose for mutagenesis look to me lower than 60 Gy. What do the authors think about this point?

Page 2, lines 85-86: The authors describe that “After irradiation, the calluses were recovered in the dark (line A recovered for 5 days, line B recovered for 12 days)”. I wonder why the calluses were recovered in the dark and why two different conditions in days were applied

Author Response

Thank you very much for reading this manuscript and providing kind comments. We have revised the parts of the manuscript that you pointed out. Moreover, you raised several questions to the manuscript, which made us think more deeply about our experimental data. We gave the explanations for these questions point to point as follows. Thank you again for your kind comments and helpful questions.

Reply 1: Thanks for your helpful suggestions. As your suggestions, we added the previous research data involved in application of 60Co-γ ray radiation mutagenesis on seashore paspalum shown in Table 1 in the introduction part. Through these known literatures, the researches about the morphological and stress tolerance variations based on callus irradiation of seashore paspalum were not found. Therefore, we aimed to construct the mutant library and evaluate their traits variations for providing excellent breeding resources and further excavating molecular mechanisms.

Reply 2: I do agree with this point. In our experiment, under the dose of 60 Gy, a large number of calluses appeared browning and vitrification, and the survival rate was only 35.69%, which was lower than the semi-lethal dose (LD50). Thus, it can be concluded that the optimal dose for callus of variety ‘Sea Spray’ is lower than 60 Gy applied in variety ‘Adalay’. However, the main purpose of this study was to obtain mutant materials. Through the morphological and stress tolerance evaluation, we found abundant variations, indicating 60 Gy dose also suitable for irradiation dosage.

Reply 3: Following our previous experiments, the induction and subculture of seashore paspalum callus needed to be performed in a dark environment. Therefore, dark condition can help for the recovery of callus after irradiation.

The institution providing irradiation is about 16 km away from our laboratory. Although the petri dishes have been sealed, considering that they have been in the environment outside the tissue culture room for a long time. We were concerned about a possible contamination of the culture medium, and then direct transfer to the differentiation medium would further extend the contamination, so we chose to recover for a period of time to observe the contamination. According to our experience, the contaminated medium usually showed obvious performance in 3-5 days, so calluses of line A were transferred to differentiation medium after 5 days.

Prolonging the recovery time of line B is based on the idea of Ye’s study [25]. Another part of Ye’s study was to subculture the irradiated callus once and then transfer it to the differentiation medium. Under this condition, at the dose of 60 Gy, the rate of shoot regeneration and plant regeneration of irradiated callus could be improved. However, she mentioned in the discussion that although a sufficient number of regenerated plants could be obtained by this method, it would inevitably increase the ratio of non-mutant plants in other species. To sum up, in view of the lower relevant data of line A and avoiding the results mentioned by Ye in discussion, we wonder whether we could improve the survival rate of callus by prolonging the recovery time appropriately, not transferring to the subculture medium. Based on the continuous observation, after recovery for 12 days, line B is higher than line A in survival rate, differentiation rate, relative differentiation rate and relative plant regeneration rate (Table 2).

Reviewer 2 Report

The manuscript presents some interesting data that will be useful to the scientific literature.

There are several issues that need revision.

The title is very general and does not convey any scientific meaning. It reads as a student thesis. Please revise accordingly.

In the Abstract, clearly state the objectives of this research study, the reasons that you undertook this study, the innovations and the significance of your results for the agronomists and plant breeders.

In the whole manuscript, the authors never state or discuss the objectives of this study. Without a clear objective, the results may be without focus.

Another issue is that the authors did not perform any phenotypic evaluation of the genetic material since everything was screened in the greenhouse. Therefore, the genetic lines that found to be tolerant to drought stress and other biotic or abiotic stresses may not be as promising when these lines are grown to field environments. I believe that the authors should discuss this.

Conclusions are short and need revision. Please state the innovations of this study, summarize your results and discuss how these results will benefit the agronomist and other scientists.

Figures and Tables are in good shape. English syntax needs improvement in certain sentences throughout the manuscript.

Author Response

Thank you very much for reading this manuscript and providing kind comments. We have revised the parts of the manuscript that you pointed out. Moreover, you raised several questions to the manuscript, which made us think more deeply about our experimental data. We gave the explanations for these questions point to point as follows. Thank you again for your kind comments and helpful questions.

Reply 1: Thanks for your suggestion. We have revised the title to “60Co-γ Ray induced Morphological and Stress Tolerance variations in Seashore Paspalum”.

Reply 2: We have revised the Abstract in manuscript following your suggestions.

Reply 3: We have added the objectives of this study in the introduction part.

Reply 4: Thanks for your suggestions. Considering the conditional consistency of light, temperature and humidity, we performed the preliminary evaluation analysis of stress tolerance under simulated stress conditions of the greenhouse. Following our previous study in grasses, we found relatively high consistency of the experiment results in the greenhouse and the field conditions. At present, all the lines have transplanted to the field and will be further observed in our ongoing work. We have supplemented the contents in the conclusions.

Reply 5: Thanks for your suggestion. We have revised the conclusions in manuscript.

Reply 6: Thanks for your suggestion. We have rechecked the manuscript and revised those English syntax mistakes.

Round 2

Reviewer 2 Report

The authors have improved their manuscript and made important revisions.

Next time, in the Author's reply, please report the line and page no. where the revisions took place when you answer a reviewer's comment.

You revised the title but I still think that it does not represent the objectives of your research. A more appropriate title would be:

Callus irradiation is an effective tool for creating new seashore paspalum germplasm for stress tolerance